# Modular and Cost-Effective Computed Tomography Design

**DOI:** 10.3390/s24082568

**Published:** 2024-04-17

**Authors:** André Bieberle, Rainer Hoffmann, Alexander Döß, Eckhard Schleicher, Uwe Hampel

**Affiliations:** 1Helmholtz-Zentrum Dresden-Rossendorf, Institute of Fluid Dynamics, Bautzner Landstraße 400, 01328 Dresden, Germany; a.doess@hzdr.de (A.D.); e.schleicher@hzdr.de (E.S.); u.hampel@hzdr.de (U.H.); 2Linde GmbH, Linde Engineering, Dr.-Carl-von-Linde-Straße 6-14, 82049 Pullach bei München, Germany; rainer.hoffmann@linde.com; 3Institute of Power Engineering, TUD Dresden University of Technology, 01062 Dresden, Germany

**Keywords:** computed tomography scanner, radiation detectors, multiphase investigations, column hydraulics, process intensification

## Abstract

We present a modular and cost-effective gamma ray computed tomography system for multiphase flow investigations in industrial apparatuses. It mainly comprises a ^137^Cs isotopic source and an in-house-assembled detector arc, with a total of 16 scintillation detectors, offering a quantum efficiency of approximately 75% and an active area of 10 × 10 mm^2^ each. The detectors are operated in pulse mode to exclude scattered gamma photons from counting by using a dual-energy discrimination stage. Flexible application of the computed tomography system, i.e., for various object sizes and densities, is provided by an elaborated detector arc design, in combination with a scanning procedure that allows for simultaneous parallel beam projection acquisition. This allows the scan time to be scaled down with the number of individual detectors. Eventually, the developed scanner successfully upgrades the existing tomography setup in the industry. Here, single pencil beam gamma ray computed tomography is already used to study hydraulics in gas–liquid contactors, with inner diameters of up to 440 mm. We demonstrate the functionality of the new system for radiographic and computed tomographic scans of DN110 and DN440 columns that are operated at varying iso-hexane/nitrogen liquid–gas flow rates.

## 1. Introduction

The intensification of mass transfer in gas–liquid contactors, like bubble columns, packed bed reactors or distillation columns, is of fundamental importance in process engineering. Internals such as structured packings are typically used in commercial column contactors to increase the surface area available for mass transfer during chemical reaction or thermal separation [1,2]. It is imperative to ensure that a uniform distribution of liquid and gas flows across the internals for the entire volume and cross-section, and, hence, a common problem of such contactors is that gas [3,4] and liquid maldistribution [5,6,7,8,9] must be tackled. The diversity of operating modes, materials and limited experimental data requires continuous investigation at the laboratory and pilot scales, and, in this regard, tomographic imaging techniques are promising but complex in terms of measurement techniques to gain insight into column fluid dynamics. Therefore, this work deals with the development of an easy-to-use and flexible imaging device based on the use of gamma radiation.

For fundamental research on phase distributions, many specialized X-ray CT systems have been introduced in the past [10,11,12]. With such systems, cross-sectional images or even three-dimensional scans with spatial resolutions of down to sub-millimeters can be obtained, with a typical scanning time of several minutes. However, owing to their limited energy, commercial X-ray systems can only penetrate a limited thickness of material, i.e., their application is limited to small columns (typically ID = 100 mm inner diameter), thin walls (i.e., low-pressure applications) and/or materials of low density. To investigate industrial columns, gamma ray CT systems with high photon energies are a better choice. Already by 1980, the design of an isotopic CT scanner for two-phase flow investigations was presented and was successfully evaluated [13]. Further CT scanners were presented over the next few decades [14,15,16,17,18,19,20,21] and were used to reveal phase distributions in large-scale devices with internals [22,23,24,25,26,27,28,29]. To obtain non-superimposed images from CT scans, radiographic projections from different angular positions from the object of investigation are required. Kak and Slaney (1988) [30] and Wang (2015) [31] provide great overviews of various computed tomography concepts and corresponding data acquisition, processing and reconstruction algorithms. Nowadays, mainly two CT scanner designs are established for radiation-based process imaging, which are described below:

First-generation scanners: Here, a *radiation pencil beam* is directed towards the object of investigation, and a single radiation detector measures the attenuated radiation. By traversing the single source and detector arrangement, a parallel beam projection is acquired. Afterwards, the arrangement is rotated in discrete angular steps around the object of investigation to acquire further parallel beam projections (see Figure 1a). After 180° rotation, the scan is complete. The obtained set of parallel beam projections is called a sinogram and is fed into computed tomography reconstruction algorithms to calculate non-superimposed cross-sectional images. This setup has its origin in simple densitometry systems and represents one of the most flexible and low-cost solutions. However, it comes with very long scanning times and experimental costs, as the studied process needs to be kept in the steady state for the entire scanning time.

Third-generation scanners: Fan beam-type CT setups, e.g., Ref. [32], use a radiation detector array (mostly an arc) and direct a *radiation fan beam* towards the object of investigation. The arc consists of many single radiation detector elements arranged with the same distance to the source spot. As the source and detector arc rotate around the object (see Figure 1b), projections are acquired until the rotation angle reaches 180° plus the fan beam angle. Subsequently, the same parallel beam sinogram data, as from the previously described single source and detector setup, are obtained. Such third-generation scanners are typically found in medical imaging. They come with fast scanning times but also higher costs as the detector must be large enough to cover the whole object with the fan beam. Thus, for industrial process tomography, which is not in daily use, an optimum between cost and scanning times must be sought, which may result in very specific and unconventional scanner designs.

As a representative application, Linde GmbH (Pullach, Germany) operates a dedicated test rig for the investigation of hydrodynamics in structured packings with organic fluids in a column of ID = 440. To observe gas–liquid phase distributions non-invasively, a pencil beam gamma ray CT system was installed in 2016 [33]. Though this setup works well, imaging times of about 8 h on average per scan are often too long for extensive measurement campaigns. Therefore, the single pencil beam scanner was upgraded to a broad multi beam scanner under some cost constraints. A further requirement is that the basic measurement principle and the corresponding analysis methods are maintained for reasons of comparability with previous measurements. With the motivation given above, the following sections describe the design of an object-adaptable CT system using a multi parallel beam projection acquisition principle.

## 2. Materials and Methods

### 2.1. Object-Adaptable CT Scanning Procedure

Due to the aforementioned cost-based issues, a simple upgrade of the existing system from a flexible but time-consuming single pencil beam arrangement [33] to a fast but fixed fan beam CT setup was not considered the best solution, as the costs for a detector arc that covers the ID = 440 column are excessive. Thus, we propose a combined CT setup as the best solution: the multi parallel beam computed tomography setup (MPB-CT). As sketched in Figure 2, a fan beam source is directed to a detector arc with a low number D of detector elements. As specified for fan beam CT setup, all detector elements provide the same distance to the source. Assuming, e.g., an odd number of detector elements D, the single pencil beam CT setup is identifiable in the center of the detector arc $D_{c}$ at angular position β, together with N further pencil beams at different angular positions β±N/2·α | N∈Z. Here, the relative angle difference α means the angle between subsequent detector elements. The detector arc to source geometry is fixed again, and the fan beam is obviously not covering the entire object of investigation. However, by moving this setup along the path s, i.e., without changing its angular position β, a set of parallel beam projections are simultaneously acquired at discrete angular positions of β±N/2·α. As this path length S can individually be adapted to both various object sizes and angular positions β, the scanning time is maximally reduced. The number of resulting detector elements $k_{M}$ for each projection angle β±N/2·α is calculated using the traverse speed ϑ, the corresponding moving length S and the detector sampling frequency f to be
(1)$$k_{M}=\frac{\vartheta }{f\cdot S}$$
and needs to be carefully selected to provide projection data with seamlessly arranged virtual parallel detectors. Therewith, the entire scanning time is reduced scalarly with the number of used detector elements D. Furthermore, the signal quality, i.e., the photon statistics at each detector position, can easily be accounted for by applying a suitable traverse speed profile in combination with a corresponding detector sampling frequency. By rotating this setup in discrete steps of $\beta_{p}=N\cdot \alpha$ around the object of investigation (maximal 180°−N·α) and by repeating projection data acquisition, a raw data matrix is acquired, from which the parallel beam sinogram data can easily be sorted/interpolated. Therefore, only the rotation axis offset $\delta_{\alpha }$ as well as the virtual parallel detector size $d_{\alpha }$ (see Figure 2) for each projection angle α need to be considered. The required movement of the detector arc and source arrangement can easily be realized with commercially available units, i.e., at least one rotating unit on which two synchronized linear units are installed.

From these projection data, non-superimposed cross-sectional images can be reconstructed using either analytical, algebraic or statistical reconstruction algorithms. Frequently, the simultaneous iterative reconstruction technique (SIRT) is used because this algebraic algorithm quickly calculates absolute attenuation coefficient distributions $\mu_{x,y}$.

Note that, in case the source to detector arc setup provides a detection angle of about $\beta_{p}=90{}^{\circ}$, only a single angular position change of 90° is required for complete CT scans. In this case, another arrangement may become interesting, in which the rotational unit—as the most challenging component to be installed onto existing industrial devices—may be completely omitted. Therefore, two identical and orthogonally arranged MPB-CT setups have to be installed.

Note also, as the traverse and rotation units are already available in the current setup, only (a) the single detector needs to be replaced with a new detector arc, and (b) the radiation pencil beam needs to be re-collimated to a corresponding fan beam. In that way, only an additional parallel beam interpolation stage as well as an innovative drive/rotating scheme needs to be added to the existing data processing unit.

### 2.2. The Gamma Ray Detector Arc

A newly developed MPB-CT scanner must maintain all features of the available pencil beam setup. Thus, the following requirements need to be fulfilled:-Maximal compatibility with the pencil beam CT infrastructure, e.g., using their existing rotating, lifting and traverse units;-High gamma photon flux and detector interaction efficiency to optimal gamma photon statistics in the shortest scanning interval;-Pulse count operation with sufficient energy resolution to perform energy discrimination for each detected gamma photon;-Insensitivity to electromagnetic fields;-Long-term stability over many hours/days/weeks;-Data transfer/detector control over long distances up to 50 m;-Power supply must be exclusively provided over the data/control interface.

With a half-life of roughly 30 years, a ^137^Cs isotopic source is selected again for the MPB-CT scanner as it delivers stable gamma photon flux over long times and photons with high energy (*E* = 662 keV) that allows for the penetration of several centimeters of steel, still producing good contrasts between gas and liquid phases. It now provides an activity of approximately A=7.4 GBq and is located inside a lead-shielding container (Figure 3) with a collimator in front (Figure 4) that limits the radiation to the fan beam to approximately 19° and a height of 25 mm. To collimate the radiation efficiently, a solid lead block with 25 mm thickness in the beam direction is used and mounted in front of the source container. The isotope capsule inside the container is operated pneumatically, and the container opening closes automatically in the case of electrical power loss.

The fan beam of the isotopic source is directed towards the radiation detector arc that comprises 16 scintillation detectors. The number of single detectors has been determined as the best compromise between an acceptable scanning time of 30 min and resulting costs. The overall detection angle of the detector arc is selected to be $\beta_{p}=180{}^{\circ}/M=16.36{}^{\circ}\;|\;M\in Z$. That means an optimal scanning procedure, as exactly M=11 discrete angular positions of the source to detector arc configuration are required for a full CT scan, i.e., to cover an object with parallel projections. Thus, the single detectors are arranged in a curved configuration with an angular distance of approximately α=1° that establishes constant detector source distance for all detector elements.

We decided to realize a made-to-measure CT scanner because the two-phase flow test rig offers unique geometries and access. Thus, the geometric adaptions of several commercially available single scintillation detectors are unavoidable. Furthermore, the direct data access of commercial detectors is usually limited, which avoids optimal integration into the test rig facility. As a scintillation material, lutetium yttrium orthosilicate (LYSO) with an active area and a scintillation light output face of 10 × 10 mm^2^, respectively, is used. This is a representative detector size for spectrometric applications found in commercially available Na(TI) detectors and, thus, for the available pencil beam setup. LYSO is selected because the light output amount is only marginally lower than that for Na(TI), but its stopping efficiency is two-times higher, the scintillation light decay time is six-times faster and it is non-hygroscopic. Together with a scintillator interaction length of 30 mm, a detection efficiency of approximately 75% for gamma photons with an energy of 662 keV is achieved. The main drawback of LYSO material is its self-activity caused by the heavy element ^176^Lu [34]. However, as long as the detected radiation flux is high enough, this background radiation can be securely subtracted. The scintillation flashlights are converted to electrical charges by the avalanche photo diode S8664-1010 (Hamamatsu), which offers a good-fitting active area of 10 × 10 mm^2^, direct coupling to the scintillation material and a very good signal-to-noise ratio because of its internal gain and insensitivity to electromagnet fields. Furthermore, Ikagawa et al. (2005) proved that these avalanche photo diodes (APDs) deliver outstanding energy resolutions in terms of a full-width at half-maximum (FWHM) of approximately 8.3% at ambient temperature of 20 °C for non-scattered ^137^Cs gamma photons [35]. However, as the internal gain of APDs is highly sensitive to temperature changes [36], a carefully designed thermal concept is developed for the final detector arc.

As shown in Figure 5, the electrical charge from the APD is initially converted to a voltage pulse using a charge-sensitive amplifier (CSA). The resulting amplitude of the voltage pulse is proportional to the energy of the interacting gamma photons. Thus, after a pulse-shaping and variable gain amplifier (VGA) stage, the pulse height value can be used to evaluate whether a scattered or non-scattered gamma photon was detected using a dual-voltage comparator (CMP) stage that realizes a simple single-channel analyzer. Because LYSO offers a very fast scintillation light decay time of 40 ns, very short pulses of approximately 1 µs in duration can be electrically shaped without any undershoot or additional pole-zero compensation stage. This also significantly reduces so-called pile-ups, which are very important when using isotopic sources with high activity, that is, a high gamma photon counting rate [37].

To provide a modular, flexible and extendable gamma ray detector arc, the aforementioned signal processing stage is assembled for four detector channels in full parallel on a single addressable (8-bit) printed circuit board. The digitized detector signals are fed into a complex programmable logic device (CPLD). Here, a 32-bit counter is implemented for each detector channel and is increased each time a voltage pulse within the predefined pulse-height range of the CMP stage is detected. Subsequent latches store the counter values for projection data read-out while the counters continue counting. The data read-out speed can be tuned according to the number of expected gamma photons per read-out interval because the counter values are transferred serially with the least significant bit (LSB) first.

Finally, four equally designed detector module boards (see Figure 5) are used to assemble the final detector arc with 16 detector channels (HZDR Innovation GmbH, Germany). All detector modules are connected in parallel to a controller board that controls the configuration of the detector modules and their data read-out. In that way, all detector channels are operated independently and simultaneously. Furthermore, it supports the bias voltage supply for the reverse operation of avalanche photo diodes and the 10/100 Mbit Ethernet communication interface to a PC. The controller board is not directly connected to a PC but to a supply board that fulfills two major tasks:-Split power over Ethernet (PoE) into (a) power supply for the detector electronic and temperature control unit and (b) standard 10/100 MBit Ethernet interface [38];-Operate the temperature control unit of the detector line.

As the IEEE standard 802.3bt-2018 [39] with power source equipment (PSE) classes of up to $P_{el}=90\;W$ is automatically identified for the PoE connection, $P_{el}=71\;W$ is available for the powered devices (PDs), that is, together for the detector arc and temperature control unit. In fact, most of the electrical power is used for temperature stabilization, which is operated independently of the detector arc. As shown in Figure 6, a Peltier element is connected to an aluminum heat sink that is internally connected to the avalanche photo diodes to maintain thermal stability. Temperature sensors that are carefully positioned at different locations along the detector arc are used to control the mode of operation (cooling/heating) of the Peltier element and the corresponding grade of cooling/heating power. A low-maintenance commercial water-cooling system effectively transfers residual heat away from the Peltier element. Finally, both the collimated source and the detector arc are mounted on the existing traverse and turntable units.

### 2.3. ATEX Test Rig

The column test rig was described in detail by [33]. The setup mainly consists of a column (ID = 440, packing bed height 1.4 m) operated with counter current gas/liquid flow (see Figure 7a). The system is operated slightly above ambient pressure (max. 10 kPa to ensure that no oxygen can enter the column) with the ambient temperature controlled to perform adiabatic tests. The test column is built of acrylic glass to allow for visual observation of the flow phenomena. In order to mimic the fluid properties of the industrial process, iso-hexane is used as liquid phase because its viscosity and surface tension are much closer to those of cryogenic liquids (see Table 1). The gas flow can be varied between 0.2 and 2.0 Pa^0.5^, while the liquid velocity can be adjusted between 5 and 20 m/h.

Unfortunately, the use of combustible hydrocarbons, i.e., iso-hexane, leads to experimental safety issues as they can form an explosive atmosphere when mixed with oxygen. The requirement for ATEX-certified [40,41] electrical components poses a problem for custom-built non-standard instrumentation, such as the MPB-CT scanner. Moreover, none of the required components, e.g., electrical motors, observation cameras, etc., are available with required certificates. Instead of undergoing an individual certification process, the entire CT setup is placed inside an acryl glass housing (Figure 7b) that is constantly purged with clean air. This setup incorporates two additional safety features:-All the moving parts are inside the acrylic glass housing. Therefore, there is no risk of becoming trapped.-The radiation that passes the detector arc is contained by a lead shield behind the detector arc that moves together with the assembly. The acryl glass housing ensures that no one can access the area between the gamma ray source and lead shield, so there is no danger of radiation exposure.

## 3. Results

### 3.1. Detector Evaluation

Initially, the signal quality of the newly assembled scintillation detectors is evaluated. Figure 8 presents a pulse-height spectrum acquired with a ^137^Cs source at an ambient temperature of 22 °C. The spectrum is acquired using the dual CMP stage. Therefore, a constant valid pulse-height range width of 64 mV is programmed with a start value at 512 mV. The constant pulse-height evaluation voltage window is then seamlessly shifted over a certain voltage range (up to 4096 mV), while each time, the number of valid gamma photons is counted for a constant scanning interval. The bias voltage of the APD is approximately −400 V. As can be seen, the photo peak at 662 keV is sufficiently developed and, thus, selectable for a valid pulse-height counting range of non-scattered gamma photons. The FWHM energy resolution is determined to be 13.5%. Using this identified non-scattered gamma photon area for the CT scans exclusively provides the best measurement accuracy. As the shape of the pulse-height spectrum is strongly influenced by the APD bias voltage and the detector operating temperature, the spectra of all detector channels must be adjusted again for the final application conditions.

As a demonstration of the temperature influence and the successfully working detector tempering design, spectra of all 16 detectors are calibrated at 20 °C using suitable APD bias voltages and the variable gain amplifier stages as fine-tuning. As can be seen in the final spectra in Figure 9 (top), a lower and upper threshold value of 1600 mV and 2900 mV can promptly be selected for the effective energy discrimination of all detector channels. Forced by a detector temperature decrease to 10 °C, the photo peak positions change to higher signal voltage ranges (Figure 9, bottom), which is due to higher internal gains of APDs at lower temperatures [36]. This, finally, proves the importance of the detector’s thermal stabilization design. In Table A1, the most important scanner properties are compiled.

### 3.2. Demonstration of Attenuation Value Evaluation

Next, the operation of the developed detector arc is demonstrated. Eventually, the liquid holdup on the packing surface is obtained from the void fraction. Initially, the CT scanner is used to evaluate the attenuation coefficient for iso-hexane $\;\mu_{IsoHex}$. Therefore, a two-dimensional radiographic scan, i.e., no multi parallel beam CT scan, is performed at a constant angular position but at different heights on a DN110 packing inserted in a beaker (Figure 10, left). This assembly is scanned once completely dry and once completely filled with iso-hexane. In total, 15 parallel projections in 10 mm height steps are obtained using the middle detector #7 only (Figure 10, right). The measured intensities for dry and wet packing ($I_{dry}$ and $I_{wet}$, respectively) are corrected by the separately measured dark counting rate ($I_{drk})$, mainly caused by ^176^Lu. According to Beer–Lamberts law [37], the attenuation coefficient for iso-hexane is calculated to be
(2)$$\mu_{IsoHex}=-\frac{1}{d\cdot a}\ln\frac{I_{wet}-I_{drk}}{I_{dry}-I_{drk}}\approx 0.0516\;{cm}^{-1}$$

The path length d is the length inside the glass beaker, and factor a is the ratio of free space to the overall volume taken up by the packing. This attenuation coefficient can then be used to quantify the liquid holdup $\bar{\varepsilon }$ with respective dry and operational (op) measurements
(3)$$\bar{\varepsilon }=-\frac{1}{\mu_{IsoHex}}\cdot \frac{1}{d\cdot a}\cdot ln\frac{I_{op}-I_{drk}}{I_{dry}-I_{drk}}$$

### 3.3. Scanning Evaluation

Next, the MPB-CT scanning procedure is quantified. Therefore, phantom cylinders (Figure 11, right/top) consisting of water, aluminum and acrylic glass are selected for scanning as these materials represent radiation attenuations within the range of different iso-hexane fractions. As mentioned above, the detector arc offers a total projection angle of 16.36° to obtain 16 parallel projections simultaneously at b=11 angular positions subsequently. For illustration, Figure 12 depicts the scanning procedure of two subsequent sweeps with correspondingly acquired detector data. While traversing the fan beam setup along the object, the detector elements are continuously sampled at positions k (see Figure 2). Afterwards, the fan beam setup is rotated by 16.36° and is traversed backwards in a way that its detector elements are sampled at the same position k. In this way, the entire phantom setup is scanned within 10 min. The fully acquired fan beam sinogram (Figure 11, left) is then re-interpolated to a parallel beam sinogram (Figure 11, middle) for non-superimposed image reconstruction (Figure 11, right/top).

Note that the obtained parallel beam sinogram dimensions match those of the previous pencil beam CT scanner. Thus, further processing steps could be used in their existing form. For image reconstruction, the simultaneous iterative reconstruction technique (SIRT) with 2000 iteration steps and an image grid of 480 × 480 pixels, that is, with a pixel resolution of 1 mm^2^, is used, as introduced by [42] and implemented in the (Python-based) ASTRA toolbox [43]. Again, by relating the measured counts in operation $I_{op}$ to the measured counts of a dry column $I_{dry}$, only the effects of liquid will become visible in the reconstruction, whereas all other static objects, such as the aluminum packing or Plexiglas column wall, will disappear.
(4)$$\mu_{x,y}=SIRT\left(-ln\left(\frac{I_{op}-I_{drk}}{I_{dry}-I_{drk}}\right)\right).$$

Finally, the liquid holdup distribution $\varepsilon_{x,y}$ is calculated by normalizing $\mu_{x,y}$ with the previously determined attenuation coefficient of iso-hexane
(5)$$\varepsilon_{x,y}=\frac{\mu_{x,y}}{\mu_{IsoHex}}.$$

Correspondingly, the averaged liquid holdup ${\bar{\varepsilon }}_{x,y}$ is calculated by
(6)$${\bar{\varepsilon }}_{x,y}=\frac{1}{N_{x}N_{y}}\cdot \sum_{N_{x}}\sum_{N_{y}}\varepsilon_{x,y}$$
using all pixels $N_{x}$ and $N_{y}$ being exclusively inside the packing column. Finally, CT scans of the phantom objects show a measurement accuracy of approximately 5% for the determination of liquid holdup and the chosen scanning interval.

### 3.4. CT Scans at the Test Rig with Iso-Hexane

Eventually, CT scans are performed for the DN440 column at different operating conditions with a shorter scanning time of 30 min. Figure 13 shows selected results of liquid holdup for the last four of the seven structured packing disks (4–7). The upper three disks are not shown here because they act mainly to uniformly distribute the inflow of the distributor.

The radiography shown in Figure 13 (left) is again from a single middle detector only, while the black line shows the averaged liquid holdup $\bar{\varepsilon }$ calculated from all 16 detector channels. The increased liquid holdup at each intersection of all disks is clearly verified in the radiographic scan. The liquid holdup shown here is calculated for every height according to Equation (3). The measurement procedure for acquiring radiograms involves moving the detector source assembly from left to right, line by line, for every desired height, without rotating the assembly. The result is a side projection of the test object. The black dots in Figure 13 (middle) represent the average liquid holdup $\bar{\varepsilon }$, which is this time calculated from CT scans (Figure 13, right) at selected packing heights and using Equation (6). The results from radiography and computed tomography provided almost the same average liquid holdup trends.

Note that the simple overview scan that is shown in Figure 13 would require many weeks with the old single pencil beam setup. The radiographic scan and the corresponding MPB-CT scans in Figure 13 are performed during a single day, which is a significant improvement in terms of the reproduction of realistic results as the column was operated continuously in steady state for the entire scanning campaign. In this way, the overall experimental costs were significantly reduced.

Figure 14 shows the CT results at the center of two disks with increasing gas flow (the liquid flow rate is kept constant). The *F*-factor is a common metric used to compare the gas flow in distillation columns, defined as $F=u\sqrt{\rho }$, where u is the superficial gas velocity and ρ is the gas density. At higher gas flows, the increasing maldistribution patterns followed the orientation of the structured packing sheets. The packing sheets are rotated 90° for every disk; therefore, the maldistribution artifacts (‘streaks’) are also 90° rotated between “Disk 5” and “Disk 6”. In addition, as expected, the total liquid holdup increased with increasing gas flow.

## 4. Discussion

We presented a new MPB-CT design for flexible tomography on industrial column contactors. The design allows for both flexible scanning modes and reduces the cost of experiments. In the presented application case, the scanning time was reduced by a factor of 16, which enables full tomographic scans on a column within 30 min and, therefore, greatly improves the quality of experimental data. Moreover, the reduced scanning time facilitates extensive measuring campaigns, because the column rig no longer needs to be operated in steady state for very long time intervals, i.e., many days or weeks. Finally, these scanning times enable acceptable human working schedules for people that have to operate the test facility.

## 5. Conclusions

In this study, a new gamma ray imaging system was presented that is perfectly suited for two-phase flow investigations in industrial environments. As an example, iso-hexane liquid distributions were successfully investigated in structured aluminum packings, with diameters of up to 440 mm and for different liquid and gas flow rates. The developed fan beam detector arc contains 16 scintillation detector elements that are operated in pulse counting mode to exclude scattered gamma photons from measurements. Its design was optimized for flexible and cost-effective industrial applications. To provide high flexibility, power supply and data interface of the scanner are established by Power-over-Ethernet.

## Figures and Tables

**Figure 1 sensors-24-02568-f001:**
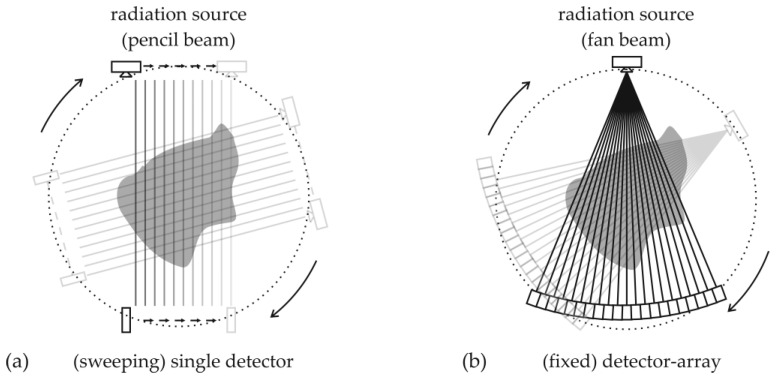
Comparison of both most common CT setups: (**a**) pencil beam versus (**b**) fan beam setup.

**Figure 2 sensors-24-02568-f002:**
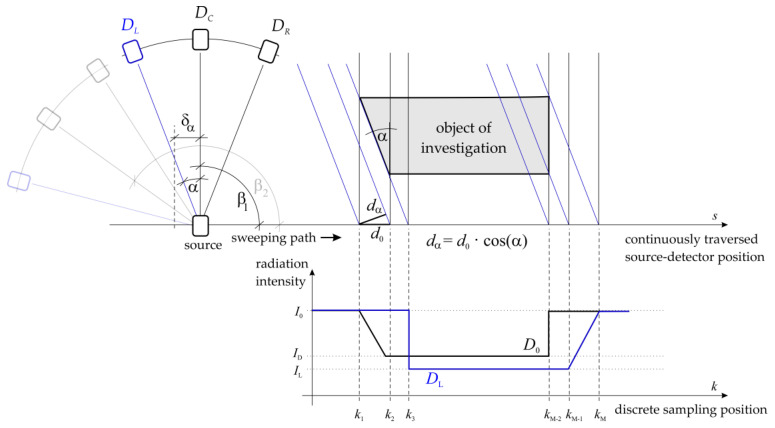
Principle sketch of an MPB-CT scanner design (index explanation: “L”—left, “C”—center, “R”—right”).

**Figure 3 sensors-24-02568-f003:**
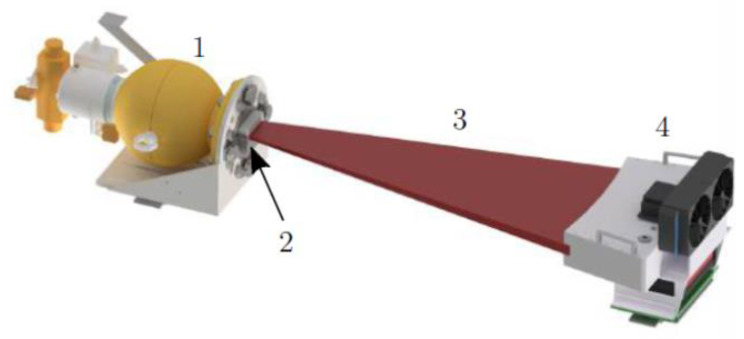
Schematic of the realized fan beam setup: (1) source container, (2) radiation collimator (details in Figure 4), (3) radiation fan beam and (4) detector arc.

**Figure 4 sensors-24-02568-f004:**
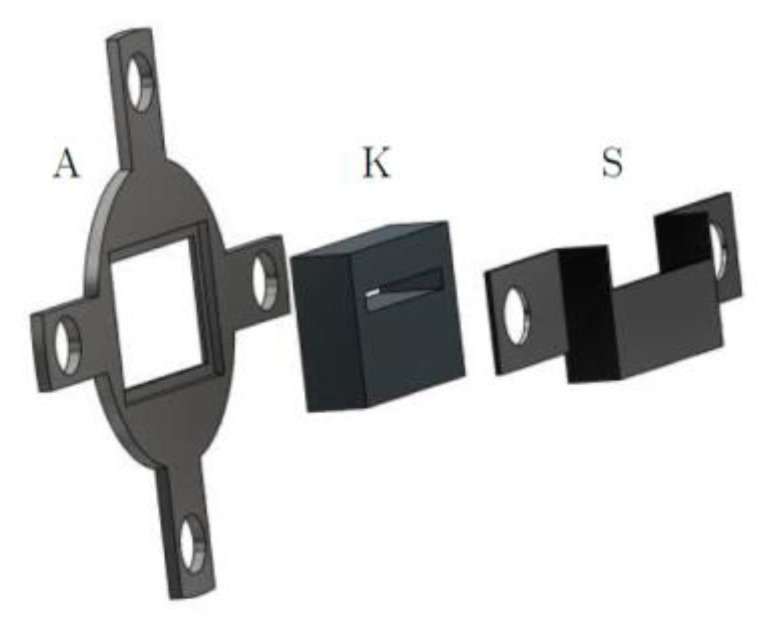
Collimator design: (A) base, (K) lead collimator and (S) holder.

**Figure 5 sensors-24-02568-f005:**
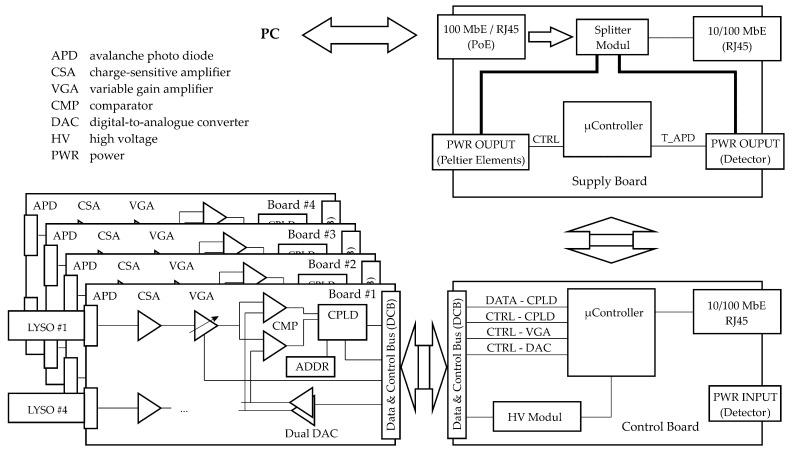
Sketch of the gamma ray detector electronics consisting mainly of four simultaneously assembled signal processing detector modules (**left bottom**), a controller board (**right bottom**) and a supply board (**right top**).

**Figure 6 sensors-24-02568-f006:**
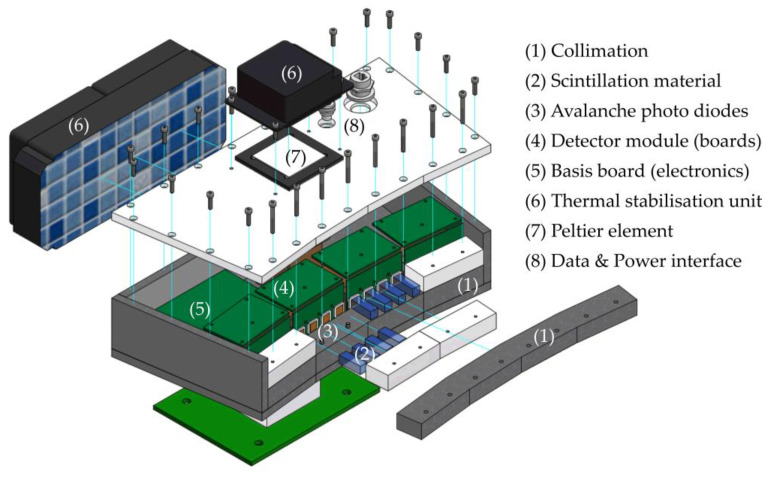
Exploded view of the fully assembled detector arc with main components.

**Figure 7 sensors-24-02568-f007:**
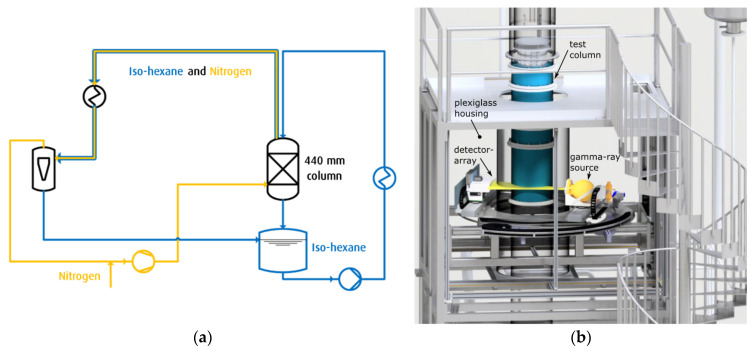
(**a**) Principal sketch and (**b**) CAD drawing of the column test rig (Linde GmbH, Pullach, Germany) for investigations on iso-hexane and nitrogen two-phase flow in aluminum packings with inner diameter of 440 mm. The drawing includes the installed MPB-CT setup that is placed inside an acryl glass housing to conform to ATEX requirements.

**Figure 8 sensors-24-02568-f008:**
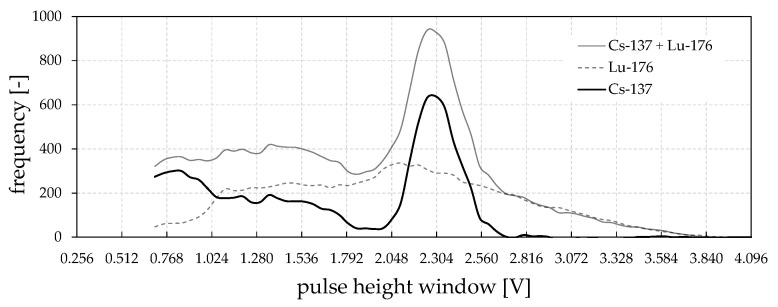
Acquired ^137^Cs pulse-height spectrum from one of the assembled scintillation (LYSO) and APD (S8664-1010) detectors at constant ambient temperature of 22 °C.

**Figure 9 sensors-24-02568-f009:**
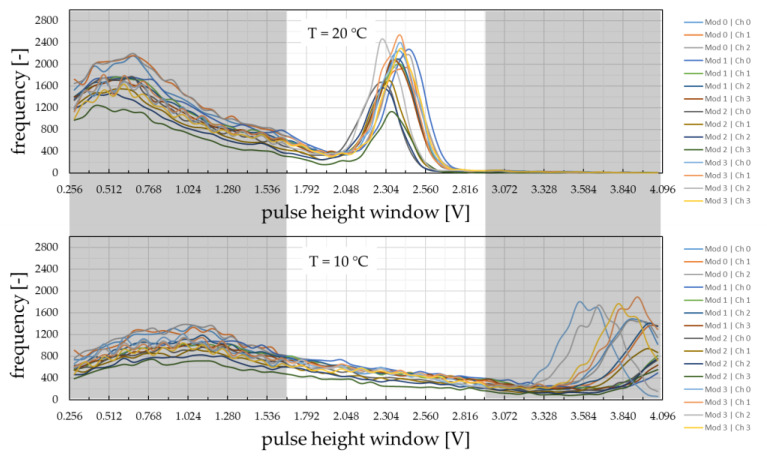
Acquired ^137^Cs pulse-height spectra for all detector channels operated at constant temperature of 20 °C (**top**) and 10 °C (**bottom**) for identical operation parameters. Grey areas represent the pulse amplitudes being excluded from counting in further measurements.

**Figure 10 sensors-24-02568-f010:**
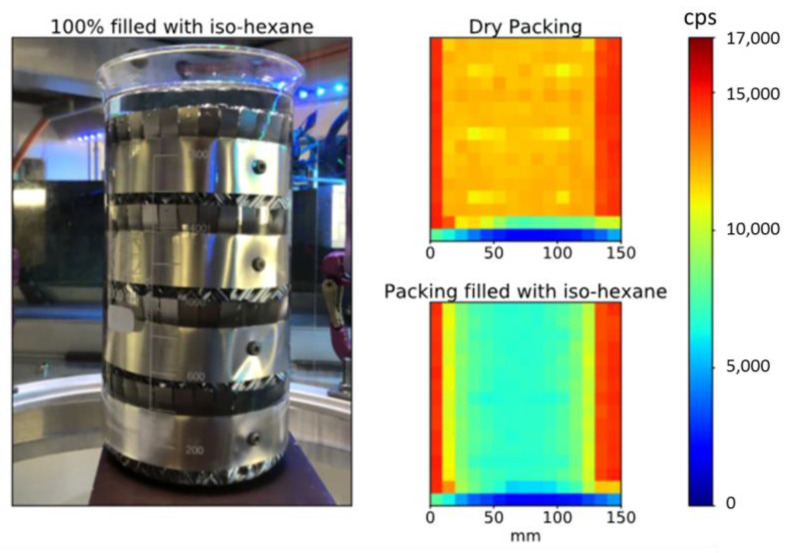
Photography of the investigated DN110 test packing (**left**) and corresponding calibration measurements, i.e., radiographic scans, in dry and liquid states (**right up**/**right bottom**).

**Figure 11 sensors-24-02568-f011:**
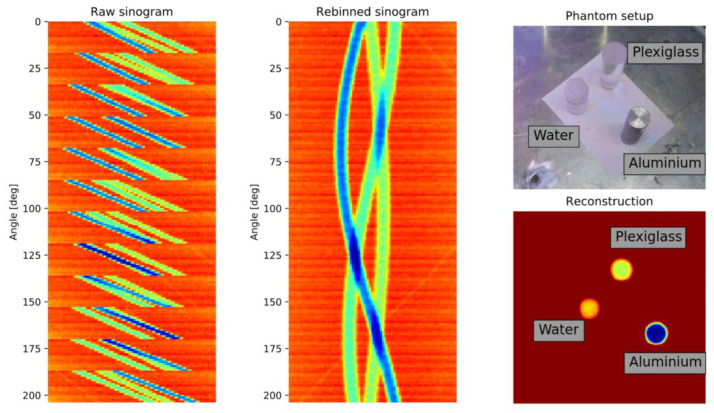
Raw data (**left**) and multi parallel beam corrected sinogram (**middle**) that are finally used to reconstruct the cross-sectional image (**right bottom**) of the objects of investigation (**right top**).

**Figure 12 sensors-24-02568-f012:**
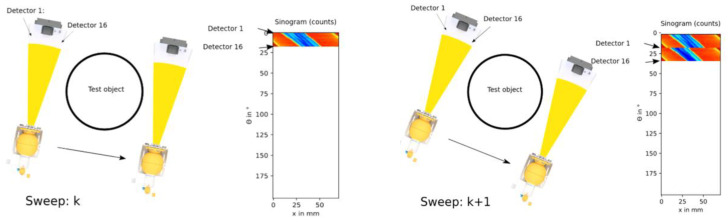
Principle schematic of the multi parallel beam CT scanning procedure.

**Figure 13 sensors-24-02568-f013:**
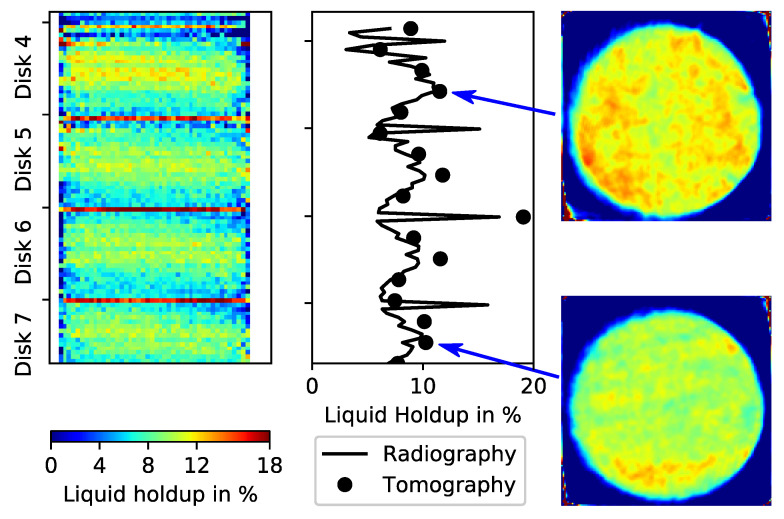
Averaged liquid holdup (**middle**) along the height of the DN440 test column using a radiographic scan (**left**) and CT scans for selected packing heights (**right**).

**Figure 14 sensors-24-02568-f014:**
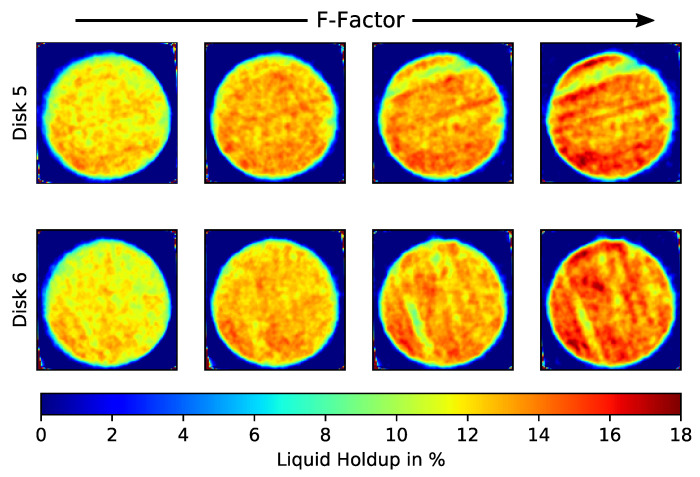
Liquid holdup distributions from the center of two different disks (5 and 6) for increasing/decreasing gas flow rates (F = 0.2, 0.4, 0.6, and 0.8 Pa^0.5^).

**Table 1 sensors-24-02568-t001:** Selected media and their corresponding properties at typical operating temperature.

Medium	Temperature (°C)	Density (kg/m^3^)	Surface Tension (mNs/m^2^)	Dynamic Viscosity (mNs/m^2^)
Liquid Argon	−185	1388	12.3	0.25
Iso-Hexane	20	653	17.6	0.31
Water	20	997	72.3	1.02

## Data Availability

Data are contained within the article.

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
