# Peer review of "Modular and Cost-Effective Computed Tomography Design"

_sensors, 2024, doi:10.3390/s24082568_

Round 1

Reviewer 1 Report

Comments and Suggestions for Authors

This is a very well-written and articulated paper. I accept this manuscript in its present format. 

Reviewer 2 Report

Comments and Suggestions for Authors

Figs. 1,2, 12 and 13 are too blurry to identify details. And Fig.1 a and Fig1 b were cited in manuscript, however there is no such label in Fig.1.

In table 1, the operating temperature of liquid Argon should not be 185 Celsius degree.  

As the Fig.10 suggestion, the measured result would be influence by temperature. Therefore, the relationship between peak position and temperature should be invested in the corrected manuscript.

The typical spectrum of Cs-137 at 25 Celsius degree would be appreciated.

Can the author tell the reader whether the equipment is functioning properly if the sample consists of heavy metals?

Figure 9 has been incorrectly labeled as Figure 1.

The pressure unit of mbar is used in this manuscript. A unit of Pa would be appreciated.

Reviewer 3 Report

Comments and Suggestions for Authors

1.      Consider revising the following lines (56-84) in the introduction to make shorter.

2.      Line 122 change statistic to statistics.

Reviewer 4 Report

Comments and Suggestions for Authors

The paper describes the development of a novel modular and economic computed tomography (CT) scanner design for multiphase flow investigations in industrial apparatuses. This type of imaging capability for larger process equipment could be quite valuable. The authors provide good technical details on the scanner hardware design, detectors, data acquisition, and scanning procedure. The functionality and validity of the scanner are demonstrated through phantom tests as well as application to a real iso-hexane/nitrogen flow system in a large column. However, the paper wasn't well organized and exsit some low-level errors.

Here are my comments to help improve the paper:

1. There were many figures too blury to read. For example, some words like "continuos transverse position" and "kM" in figure 2 are hard to read. The figure 12 and figure 13 are also very blury.

2. In line 11, the "overall" should be "a total of"

3. In section 2.1, the authors introduced a CT scanning procedure. But the corrsponding  reconstruction method was discussed very briefly. More mathematicaldetails could be provided on the CT reconstruction algorithm.

4. The figure "Acquired 137Cs pulse height spectrum from one of the assembled scintillation (LYSO) and APD (S8664-1010) detectors at constant ambient temperature of 22 °C." should be figure 9 instead of figure 1.

5. The authors mentioned the detector has a energy resolution of 13.5%. Is this energy resolution only used for evaluating whether the gamma photon was scattered or non-scattered?

6. The authors mentioned the final detector arc has 16 detector channels. Can these channels collects signals simultaneously or they work in sequence?

7. The authors said therefore, there is no risk of becoming trapped because all the moving parts are inside the acrylic glass housing. Is there a chance of the components to gett trapped with their cables?

Comments on the Quality of English Language

The Quality of English Language was good overall.

Round 2

Reviewer 4 Report

Comments and Suggestions for Authors

All my comments have been addressed very well in the revision. I just have one more question.

I know that the proposed gamma ray CT scanner is designed for industry use, the accumulated radiation dose is not a main concern. But have the authors done any simulation studies on radiation dose and the tube current exposure time using software like GATE?

Author Response

Thank you for that question. We have not performed such simulations as the effort to model all conditions of/in the laboratory correctly and completely is too high compared to its benefit.

We, instead, bordered the isotopic source to a defined fan beam of certain height/angle and added lead material behind the detector arc to efficiently absorb residual radiation that still passes the detector. Furthermore, scattered radiation within the object of investigation is (in our case) neglectable as the object itself absorbs it.  

In a final step, we mapped the generated radiation field in the laboratory using standard dosimeters to be sure to be below a specified dose. Otherwise, the permission would not be given by the corresponding  agency.